# Dynamic Mechanical Analysis on Delaminated Flax Fiber Reinforced Composites

**DOI:** 10.3390/ma12162559

**Published:** 2019-08-11

**Authors:** Yiou Shen, Jiayi Tan, Luis Fernandes, Zehua Qu, Yan Li

**Affiliations:** 1School of Aerospace Engineering and Applied Mechanics, Tongji University, 1239 Siping Road, Shanghai 200092, China; 2Aerospace Engineering, Instituto Superior Técnico, Av. Rovisco Pais, 1, 1049001 Lisbon, Portugal; 3State Key Laboratory of Molecular Engineering of Polymers, Department of Macromolecular Science, Fudan University, 2005 Songhu Road, Shanghai 200433, China

**Keywords:** dynamic mechanical analysis, delamination, plant fiber-reinforced composite, damping properties

## Abstract

It is well-known that the presence of the delamination in a plant fiber-reinforced composite is difficult to detect. However, the delamination introduces a local flexibility, which changes the dynamic characteristics of the composite structure. This paper presents a new methodology for composite laminate delamination detection, which is based on dynamic mechanical analysis. A noticeable delamination-induced storage modulus reduction and loss factor enhancement have been observed when the delaminated laminate was subjected to a forced oscillation compared to the intact composite laminate. For delamination area of 12.8% of the whole area of the composite laminate, loss factor of approximately 12% increase was observed. For near-to-surface delamination position, loss factor of approximately an 18% increment was observed. The results indicate that the delamination can be reliably detected with this method, and delamination position shows greater influence on the loss factor than that of the delamination size. Further investigations on different frequencies and amplitudes configurations show that the variation of loss factor is more apparently with low frequency as well as the low amplitude.

## 1. Introduction

Plant fiber reflects outstanding and comparable mechanical and dynamic mechanical properties to synthetic fiber such as fiber glass, this leading to extend its applications for special engineering materials such as aerospace and automotive interior parts, and construction structures [1,2]. Although the specific mechanical properties of plant fiber are comparable to the synthetic fiber, their polymer matrix-based composites exhibit inferior mechanical properties. One of the primary reasons is the weak interfacial bonding between the fiber and matrix, and this leads to a low efficiency of stress transfer which limiting the mechanical performance of composite. Moreover, this weak interphase causes matrix crack and delamination more easily than artificial fibers during composite manufacture and service, this leads to substantial degradation in structural stiffness and strength. Since the mechanical properties such as flexural, shear and impact properties are the first priority for the component structure design, the increasing use of plant fiber-reinforced polymer (PFRP) composite in load-bearing structural applications make it is urgent to find effective ways to detect the defects and delamination within the PFRP composite.

Delamination is a common damage mode in laminated composites, which is occurring due to the low interlaminar strengths of the FRP laminates [3]. It can be generated due to manufacturing defects, low-velocity impact, lightning and so on, which could resulting substantially decline on compressive residual strength and structural integrity [4,5]. Since delamination is an internal damage, the opaque inherent characteristics of plant fiber make it not visible from the outside. Many Non-Destructive Inspection (NDI) techniques have focus on characterization defects within the plant fiber-reinforced composite such as ultrasonic [6,7,8], acoustic emission [9,10], thermography [11] and so on. However, these conventional methods are often expensive and depend on the skill and experience of the operator. It was found that delamination within laminated composites could not only cause local discontinuities in structural stiffness but also have influence on damping properties which can affect the dynamic characteristics and modal parameters (natural frequencies) of structure [12]. The frequency variation between intact and damage status of composite materials enables the study of potential damage in composite laminate structures. Junker et al. [13] found damage and resulting decrease in matrix stiffness can lead to significant changes in the overall, frequency-dependent damping and dynamic stiffness of the composite under time–harmonic two-dimensional loading. Lai and Young [12] found interlaminate delamination cause the decrease of modal frequency and increase of modal damping in graphite/epoxy composites. Zhang et al. [14,15] investigated the size and location parameters of delamination in carbon fiber-reinforced plastic (CFRP) and glass fiber-reinforced plastic (GFRP) plates based on the changes in natural frequencies and these parameters were predicted using a Surrogate Assisted Optimization method. Yang and Oyadiji [16] found the behavior of the modal frequency deviation depends on the axial location as well as the interlayer position of the delamination.

Dynamic mechanical analysis (DMA) is a powerful technique on characterizing not only molecular motion, but also phase morphology of polymers and reinforcements, it provides a sensitive and nondestructive detection of the interfacial region within laminated composites [17,18]. The DMA machine can measure two types of elastic (stiffness) and damping (energy dissipation) response to a low-strain periodic deformation. In composite structures, the molecular motions at the interfacial region contribute to the damping of materials. The molecular damping magnitude can be used to quantify the interface bonding of composite. Therefore, DMA has been applied to composite interface characterization [19,20,21]. The dynamic mechanical properties of fiber-reinforced composites including storage modulus (*E′*), loss modulus (*E″*) and loss factor (tanδ) etc. are temperature dependent, and these parameters are usually influenced by dynamic loading conditions. For example, in aerospace and transportation fields, the engine noise, airflow and bump during turbulence will cause various vibration conditions. Moreover, the dynamic mechanical response of composites also depends on the physical or structural arrangement of phases such as interface, morphology and the nature of constituents. Researchers elaborated that the effect of the presence of compatibilizer, fiber content, fiber size, fiber orientation, stacking sequence, fiber/matrix interface, fiber breakage, matrix cracking, hybridization governed the dynamic mechanical properties of a composite material [22,23,24,25,26,27]. Saba et al. [18] and Gupta et al. [28] summarized the researches on dynamic mechanical property of natural fiber-reinforced polymer composites for last decades, and provided valuable information for further investigations and in the elaborative application of DMA for evaluating the natural fiber-reinforced polymeric composites.

As mentioned previously, delamination could seriously influence the structural integrity of composite and cause fateful consequences, the impact of delamination on dynamic mechanical properties of composite need a more in-depth research. Since delamination within composite laminate could also have influence on damping properties which can affect the dynamic properties of structure, DMA could be used to characteristic delamination of the composite. However, this method has not been used in characterization delamination so far. Some papers mentioned above are regarding damping properties of delaminated composite plates, but most of them focus on the natural frequency behavior of such composites and do not assess the dynamic mechanical properties, which are the focus of this study. Hence, the effect of delamination with various size and transverse position on flax fiber-reinforced polymer (FFRP) composite laminate with regard to temperature, frequencies and amplitude are evaluated using DMA in this study. As such, this paper provides a starting step towards a better understanding of the viscoelastic behavior of delaminated natural fiber composites.

## 2. Materials and Methods

### 2.1. Materials and Fabrication

Unidirectional (UD) flax fabric reinforced epoxy resin composite laminates were manufactured in this study. The unidirectional flax fabric has average thickness and areal density of 0.2 mm and 200 g/m^2^, respectively, and it was provided by Lineo Co. Ltd., MeulebekeBelgium. Epoxy resin (NPEL–128), amine curing agent (EH–6303), and accelerator (EH–6412) (100:26:2 weight ratio) were supplied by Nanya Electronic Materials Co., Ltd., Kunshan, China.

Prior to manufacturing, the fabrics were dried in an oven for 4 h at 120 °C in order to remove moisture. A single layer of PTFE (Teflon) film (15 µm in thickness) was placed at the pre-defined locations during layup to introduce delamination [15]. The composite laminate consisted of six layers of UD flax fabrics, three type of delamination areas at the same transverse position (mid-plane) was investigated, this included 100 mm^2^, 25 mm^2^ and 6.25 mm^2^. Moreover, delamination located in three different transverse positions was also investigated, including Position I (at middle thickness), position II (at 1/3 thickness) and position III (at 1/6 thickness). Square Teflon films were carefully stitched at two cross corners to one layer of the flax fabric at the correct location with a fine cotton thread in order to prevent moving positions during the manufacturing process. Figure 1 schematically shows the positions where the delamination was introduced in the samples.

Vacuum-Assisted Resin Infusion (VARI) technique was used to fabricate composite laminates as shown in Figure 2. The flax fabric layup was placed on the top of a smooth glass table between two sheets of peel plies. The resin system was degassed in a vacuum oven for 30 min to remove air bubbles before perfusion. Once the vacuum was surely set, the resin was impregnated in the mold and the flow was evenly distributed across the plate. The laminates were then cured at room temperature for 24 h and post-cured at 120 °C for 2 h. The fiber weight fractions were approximately 40% for all the specimens and the porosity of the laminate was less than 1%, which was inspected with C-scanning machine. Three samples were carefully cut from the laminate and polished to the same size for each design in order to obtain relatively accurate values. The samples tested in this study are summarized in Table 1. Figure 3 shows the cross-section microscopic photograph of the sample and it reveals the introduced delamination. This micrograph captures the hierarchical structure of flax fabric and confirms that the delamination has been successfully introduced.

### 2.2. Testing and Data Processing

Dynamic Mechanical Analysis (DMA) is used to study the physical properties of a specimen subjecting to a small sinusoidal oscillating force as a function of time, temperature, and frequency. Due to the viscoelastic properties of polymer-based composite samples, the two signals (stress and strain) will be out-of-phase, and a phase shift δ can be measured. The complex modulus *E** is given by the ratio of the stress amplitude to the strain amplitude. The complex modulus is composed of the storage modulus (*E′*) and the loss modulus (*E″*). *E′* represents the stiffness of a viscoelastic material and is proportional to the energy stored during one loading cycle. *E″* is proportional to the energy dissipated in one loading cycle. These two moduli can be calculated according to Equations (1) and (2) [17]:(1)$$E^{\prime }=E^{*}\cos\delta$$
(2)$$E^{\prime \prime }=E^{*}\sin\delta$$

Then, the complex modulus can be calculated according to Equations (1) and (2):(3)$$E^{*}=\sqrt{E^{{}^{\prime }2}+E^{{}^{\prime \prime }2}}$$

The complex shear modulus can be calculated according to the complex modulus, and given as:(4)$$G^{*}=\frac{E^{*}}{2\left(1+\nu \right)}$$
where the Poisson’s ratio ν = 0.3.

It should be noticed that the *E′*, *E″*, *E** and *G** are dependent on the sample dimension. Therefore, two geometry independent parameters tanδ and *G″* are used to characterize the delamination within the composites. The loss factor (tanδ) is a dimensionless parameter representing the energy lost per cycle expressed in terms of recoverable energy, and gives a measure of the damping or internal friction in a viscoelastic system which can be obtained by Equation (3):(5)$$\tan\delta =\frac{E^{\prime \prime }}{E^{\prime }}$$

It was indicated by Kennedy et al. [20] that the loss modulus, *G″*, is related to the viscous dissipation as the energy dissipated per cycle per unit volume of the sample, and it gives better characterization on the effect of the fiber/matrix interface. The loss shear modulus, *G″*, can be expressed in terms of *G** and δ as:(6)$$G^{\prime \prime }=\left|G^{*}\right|\sin\delta$$

The dynamic mechanical properties of the composites were obtained using the Q800 DMA machine (TA Instruments, New Castle, DE, USA) according to ISO 6721–1 [29] in a dual cantilever mode as illustrated in Figure 4. All the samples were tested in two scanning manners, this included temperature scans and time scans. The frequency and amplitude configurations for time scans were also investigated. The test parameters for these three manners are detailed in Table 2.

As for the temperature scans manner, firstly, the sample was clamped onto the fixture and a temperature scanning between 25 and 120 °C was realized with a ramp rate of 5 °C/min, the frequency and amplitude were set as 10 Hz and 10 μm, respectively. For the time scans, firstly, four different frequencies were tested for each sample, namely 1, 10, 50 and 10 Hz, and all tests were conducted at a fixed strain amplitude of 10 µm. Ambient temperature of 25 °C was chosen as the testing temperature for all samples and isothermally for 5 min. Secondly, five different strain amplitudes were tested for each sample, namely 10, 15, 20, 25 and 30 µm and all tests were conducted at a fixed frequency of 10 Hz. The samples were equilibrated at 25 °C for 5 min.

The dynamic mechanical properties data of the composites obtained in frequency and amplitude scan modes were processed using the Matlab^®^ (MathWorks, Natick, MA, USA) in accordance with the following steps: Firstly, since the total recorded testing time for each sample was 5 min, the loss factor results were averaged to obtain significant results. Secondly, the average loss factor value in five minutes duration of three samples for each delamination area or position were averaged again. Figure 5 illustrates the averaging process for the loss factor results of DEL_2.5_I (delamination area 6.25 mm^2^). A total of three signals were obtained from three samples produced with the same delamination area of 6.25 mm^2^. Dash lines with white markers are variations of loss factors against time for three samples, these signals were subsequently averaged in time, as represented by the dashed lines with black markers. The final loss factor of the specimen DEL_2.5_I was obtained by averaging the above black marker values and represented as a black line in Figure 5, where the blue line is the error bar. This process was conducted for each of the specimens and other properties obtained from the DMA machine, such as the storage modulus and the loss modulus.

## 3. Results

### 3.1. Influence of the Delamination Area

#### 3.1.1. Temperature Scans

Figure 6a shows the typical plots of storage modulus and loss modulus against temperature for the FFRC specimens with various areas of delamination. It can be seen from the graph that the intact composite exhibits slightly higher storage values than that of other delaminated composites with the increase of temperature due to the reason of storage modulus reflects the inherent elastic modulus of the laminate. The prefabricated delamination destroys the structural integrity of the composite laminates. In addition, the storage modulus of the FFRP laminates with different delamination areas show similar values and trends. It can be seen from the graph that the peak loss modulus of the DEL_5_I sample show the highest value, and the intact laminate shows the lowest value as expected. This indicates that the delamination rises the energy loss in composite internal motion when being applied a dynamic stress. The complex modulus and complex shear modulus are then calculated according to the Equations (3) and (4), these data were plotted against temperature as shown in Figure 6b. Clearly, the intact composite laminate shows higher complex modulus and complex shear modulus than that of other delaminated samples.

In dynamic mechanical testing, tanδ is also referred to damping and is an indicator of how efficiently the material loses energy to molecular rearrangements and internal friction. The magnitude of the loss shear modulus directly reflects the energy dissipated during testing. In a composite material both the matrix polymer and the interfacial contribute to these losses. In this study, the matrix and the fiber/matrix interfacial properties are the same. Therefore, the characterizing on the interlayer bonding or delamination performance depends on the relative magnitude of the interlayer contribution to the overall dynamic properties of the composite specimen. Therefore, when composite materials are subjected to dynamic mechanical testing, any decrease in the strength of the interlayer bonding should be reflected by an increase in the measured values of both tanδ and *G″.* The variation of tanδ and *G″* were calculated according to Equations (5) and (6), these data were then plotted in Figure 7a. It can be seen from the curves that the intact laminate (NODEL) displays the lowest tanδ and *G″* due to its integrated interlaminar interface. This seems reasonable, the interlaminar interface of the intact laminate are assumed to be well bonded, whereas, the delaminated zone of laminate is assumed to yield a very weak interlaminar interfacial bond, and this causes the extra energy dissipation during forced vibration of a DMA test. It can be seen from Figure 7b that the different sized preset layers inside of the laminate causes the variation on both peak value of tanδ and *G″*. The horizontal axis on each plot presents the delamination area both in SI units (mm^2^) and in terms of percentage area of the whole specimen. In another word, the 100 mm^2^ delamination area corresponds to a 12.8% delaminated area for the DEL_10_I specimen. It can be found from Figure 7b that the value of tanδ and *G″* increases with the proportion of delamination area to the whole area of the sample. In terms of the 3.2% (DEL_5_I) delaminated sample, both of these two values raise approximately 6% than that of the intact laminates. However, when the delamination area increased to 12.8% (DEL_10_I) of the whole area of the sample, the tanδ and *G″* reduces to the same value of the intact laminate. This indicates that small delamination area can causes the enhancement of the loss factor and loss shear modulus due to the friction between adjacent separate layers in delaminated zone during forced vibration. However, if the delamination area is too big, the energy consumed through internal friction and molecule motion within the FFRP composite are seriously reduced, which cause a decrease in total energy dissipation.

#### 3.1.2. Frequency Scans

Figure 8a presents the variations of *E′* and *E″* with delamination areas tested at a fixed strain amplitude of 10 μm and frequencies of 1, 10, 50 and 100 Hz, respectively. It can be seen from the graph that the storage modulus of the FFRP samples decreases linearly (dash line) with the enlargement of the delamination area at four different test frequencies. The *E′* of the largest delamination area sample (DEL_10_I) is approximately 15% lower than that of the intact one for all four test frequencies. This is due to the reason that bonds between fiber/resin and resin/resin in the composite provide energy storage capabilities during deformation. However, the introduced delamination prevents the resin/resin bonds adjacent this area from forming. The discontinuous interface in the delamination area can be assumed has a very week interfacial adhesion between two layers, this translates to a lower capacity of the laminate to store elastic deformation energy during vibratory oscillation. As a result, the specimen presents a lower storage modulus when compared to the intact one. Li et al. [30] also found the Young’s modulus of the sandwich beam decreased with the delamination length increase due to the stiffness reduction. It can also be found that samples with same delamination areas exhibit similar *E′* values at 10, 50 and 100 Hz test frequencies, which are slightly higher than that of the 1 Hz testing results. Similarly, the loss modulus was plotted as a function of the delamination area in Figure 8a. As presented in the temperature scans, the variation of loss modulus with delamination area is not significant between the delaminated and intact sample, slightly enhancement was observed on *E”* for small delamination area, and the value decrease with the delamination area keep increase. However, there is an approximately 45% difference on loss modulus between test frequencies of 1 Hz and 100 Hz for all test sample. This indicates that the energy loss for this type of composite exhibits a dependence on the test frequency instead of delamination area. Dong and Gauvin et al. [21] also found increase the test frequency will significantly decrease the *E″* of a CFRP laminate during a DMA test, whereas, the effect on *E′* is not observed.

The variation of complex modulus and complex shear modulus with delamination area is shown in Figure 8b. It can be found that both *E** and *G** decrease with the enlargement of the delamination area. The reductions for both *E** and *G** of the largest delamination area composite (DEL_10_I) are approximately 10% compares to the intact laminates. The values of *E** and *G** are dependent on both of *E′* and *E″*. Therefore, *E** and *G** also decrease linearly with the increase of delamination area.

The variation of the overall loss shear modulus and loss factor against delamination area is shown in Figure 8c. It was found that the smallest delaminated samples (DEL_2.5_I, 1%) show the highest *G″* value and it was 5% higher than that of the intact laminates tested at 1 Hz. After that, the *G″* decreases steady with the increase of delamination area, and the value of the largest delaminated sample (DEL_10_I, 12.8%) was 5% lower than the intact laminates. There is no significant increase on the *G″* for the small size delaminated laminates at other test frequencies. This result indicates that small size delamination at low test frequency causes slightly increase on *G”* but not for large size delamination samples and higher test frequencies. The average loss shear modulus of the specimens tested at low frequency is much higher than samples tested at higher frequencies, and this difference is as significant as 50%. The loss factor of the largest delamination area (DEL_10_I, 12.8%) is approximately 12% higher than that of the intact one for lower test frequencies of 1 Hz and 10 Hz. However, this difference become insignificant with the test frequency increase, and it was found the loss factor for the delaminated and intact sample are similar.

#### 3.1.3. Amplitude Scans

It can be found from Figure 9a that the storage modulus tested at different strain amplitudes shows same reduction trend to that of tested at different frequencies. Clearly, the effect of the amplitude on *E′* is not essential, all the storage modulus for the same laminate tested at different amplitudes falls to the same point in the graph. Again, the delamination area does not show great influence on the loss modulus. However, the loss modulus increases with the strain amplitudes, approximately 30% enhancement was observed for strain amplitudes of 30 μm compared to that of tested at 10 μm. The increase of *E″* with amplitude observed for the composites was attributed to the Coulomb friction at the interfaces due to the reason that high amplitude would yield a more significant loss signal [30]. In addition, the complex modulus and complex shear modulus decrease linearly as delamination increases, and there is an approximately 13% reduction on both of the values as shown in Figure 9b. As for the loss shear modulus, increase the strain amplitude results in an enhancement of *G″* for all delaminated and intact laminates and a maximum reduction of 8% was observed on the largest delaminated laminates (DEL_10_I, 12.8%) tested at the 30 μm strain amplitude compares to the non-delaminated ones. In another hand, the loss factor also growth with the delamination area as stated in Section 3.1.2 as shown in Figure 9c. Moreover, the strain amplitude has great effect on the lost factor, the approximate increment is 30% for all the samples when the amplitude is 30 μm. When the amplitude increases, the internal deformation and friction of the molecule and fibers will increase, and the dissipated energy will inevitably increase. Therefore, the material loss factor will increase with the increase of the amplitude.

### 3.2. Influence of Transverse Delamination Position

#### 3.2.1. Temperature Effect

The dynamic properties of three different specimens with the same delamination area (25 mm^2^) at different transverse positions are illustrated in Figure 10 and Figure 11. Figure 10 presents the typical *E′, E″*, *E** and *G** curves against temperature of FFRP with same delamination area but different transverse positions. The different delamination positions investigated in this study are schematically represented in Figure 1. It can be found that the divergence on laminates delaminated at varied interlayers is quite apparent. The intact composite shows the highest *E′*, *E** and *G**, and the value of these three parameters decrease with the delamination far from the midplane. However, the *E″* shows the opposite trend. Similarly, the typical *G″* and tanδ against temperature curves in Figure 11a suggest that these two parameters show great enhancement for laminate with delamination close to the surface. The peak value of the loss shear modulus and loss factor are summarized in Figure 11b. The top horizontal axis shows the transverse position in the non-dimensional form of *2x/t*, where *x* is measured from the mid-thickness of the specimen outwards (as indicated in the plot) and *t* is the specimen thickness. The bottom horizontal axis provides the correspondence between the nomenclature used and the transverse position of the preset delamination. It was found that the peak value of *G″* for laminates with different delamination transverse positions are similar, which is 6% higher than that of the intact laminate. This indicates that transverse position of the delamination does not affect the total amount of energy dissipated caused by delamination and the value is consistent with the G″ value of middle-size delaminated sample. However, the loss factor tends to increase with the transverse position moves from mid-plane (*2x/t =* 0) to a near-surface delamination. In this study, the maximum value of *2x/t* is 0.67, which is corresponded to Position III. In this transverse position, the delamination caused the highest enhancement on peak value of tanδ, which is 25%. This suggests that laminates with delamination in different transverse position exhibit distinct energy dissipation efficiency and damping properties.

#### 3.2.2. Frequency Effect

Figure 12a shows the storage modulus *E′* tested at a fixed amplitude of 10 µm and frequencies of 1, 10, 50 and 100 Hz as a function of the transverse delamination position. A 15% decrease in storage modulus is observed for delamination position changing from mid-plane to the near-surface of the sample at all four frequencies. As a result, the experimental storage modulus values fit a 2nd order polynomial which is different with the linearly downward trend of delamination area variation. It can be seen in the same figure that the loss modulus of positions II and III descend approximately 10% than that of the non-delamination laminate and laminate with delamination in position I at 1, 10 and 50 Hz. However, the effect of transverse position on delamination is not significant when test under 100 Hz. Again, the test frequency has a remarkable influence on the loss modulus of the FFRC laminate, the difference between the value of all four type of FFRC laminate tested at 1 and 100 Hz are approximately 45%. Clearly, the delamination transverse position also has a stronger effect in the *E** and *G** of the FFRP composites as shown in Figure 12b, and the decline trend curve as shown in the dash line also fits a 2nd order polynomial. Laminates delaminated at positions II and III show similar value on *E** and *G**, which is 15% lower than that of the intact ones. In Figure 12c, the loss shear modulus shows similar trend to the loss modulus that delamination transverse position has apparent influence under lower tested frequencies. The loss factors from the specimens with delamination at positions II and III shows an increase of over 18% compared to that of the non-delamination specimen. In addition, increase the test frequency causes reduction on the loss factor for all the samples, the maximum reduction can be 50%.

#### 3.2.3. Amplitude Effect

The dynamic properties of FFRP laminates varied with delamination transverse position which can be tested under a fixed frequency of 10 Hz and five strain amplitudes of 10, 15, 20, 25 and 30 μm are shown in Figure 13. Both of the decline trend of storage modulus and loss modulus for five set of results tested under different amplitudes follow 2nd order polynomial curves as the dash lines shown in Figure 13a. The laminate with delamination at transverse position of *2x/t* = 0.34 (Position II) have the lowest storage modulus and loss modulus, which are approximately 15% lower than that of the intact laminate. Compared to the storage modulus, the strain amplitude has a greater impact on the loss modulus, a 30% growth was observed between laminate tested at 30 μm and 10 μm strain amplitude. The same reduction trend was found for the *E** and *G**, the maximum decrement is 15% for laminate with delamination at transverse position of *2x/t* = 0.34 (Position II). As for the *G″* and tanδ, it was found that raise the amplitude lead to an enhancement of at least 20% on the loss shear modulus and loss factor of FFRP laminates. The reduction caused by delamination transverse position is approximately 15% and 3% for *G″* and tanδ, respectively.

## 4. Discussion

The impact of delamination size and transverse location on the dynamic mechanical properties are the two main concerns in this study. Firstly, in terms of delamination size, the storage modulus decreases with the delamination size increase as expected. However, this does not mean the loss modulus goes the opposite way. Actually, the energy dissipation through friction and molecule motion will go down if the delamination size is too big. In fiber-reinforced composite material, viscoelastic damping appears to be the most dominant mechanism in undamaged polymer composites vibrating at a small amplitude. The interface phase is a region with a certain thickness on the surface and nearby layers of the composite material, it processes different properties to both fiber and matrix. As a bridge connecting adjacent layer, interlaminar interface phase has a great influence on the overall performance of composite materials. When stimulated by external excitation, the interface will produce a large shear strain with the deformation of the material, and these shear strains will cause friction within the material and dissipate the energy. The delamination equivalent to lower the interlaminar bonding and this leads to extra friction and local flexibility in the interlaminar debonding region, and this results in a higher loss factor. Moreover, it seems that the test frequency has remarkable effect on *E″*, *G″* and tanδ, which means test frequency has great influence on energy dissipation of composite materials. Some researchers have reported that delamination induced modal frequency shifting and modal damping increasing due to modal stiffness reduction [12,16,31].

Secondly, in terms of delamination transverse position, the dynamic mechanical properties are quite distinct in spite of all three laminates process same delamination size. It was found that among the three interlaminar position, mid-plane delamination laminate provides the highest dynamic mechanical properties, while other two delamination position laminate show similar properties. Also, it was found that the transverse position of the delamination has a greater effect on the damping parameter (tanδ). Mei et al. [32] also found that the different sizes and transverse position of delamination within a carbon fiber-reinforced polymer (CFRP) composite significantly modified the local defect resonance frequency based on a vibration in-situ detection method. They found strong local vibrations only occurred in the delamination regions. Therefore, the delamination could be successfully detected. Their results also proved that the effect of delamination depth (transverse position) on the detectability of the delamination was more significant than the size of the delamination, which is same to the conclusion in this study. Thus, dynamic mechanical analysis can and with considerable certainty, detect the delamination within the composite samples. Moreover, this technique is easy to operate. During a DMA delamination detection process, a control sample without any defect should be prepared, the increase on the loss factor indicates the existence of delamination in test sample with the same manufacture and structural configurations to the control sample. Considering the extensive application of DMA in industrial field, from polymer industries to aerospace industries [18], and hence this method is not expensive.

For future work, it is desirable that the methodology should be extended to detect and quantify the size and axial/transverse location of the delamination in complex composite structures not just for unidirectional composite.

## 5. Conclusions

In this paper, unidirectional flax fiber-reinforced composite laminate with different delamination areas and transverse positions were implemented using the dynamic mechanical analysis machine under temperature, frequency, and amplitude scanning manners. It was found that the delamination of different sizes and locations significantly affected the dynamic mechanical properties of the composite. The delamination could be reliably detected using test parameters with relatively low frequency and low amplitude, the increase on loss factor indicates the existence of delamination. Moreover, experimental results indicated that the effect of delamination transverse position on the detectability of the delamination was more notable than the size of the delamination. It was also found that the frequency has greater influence on energy dissipation of composite compares to the effect caused by amplitude. In summary, despite this, the DMA method requires more data obtained from a large number of delamination sizes and positions to get empirical formula, which might quantify the delamination parameters. This methodology has been proven useful for the detection of delamination in unidirectional composite.

## Figures and Tables

**Figure 1 materials-12-02559-f001:**
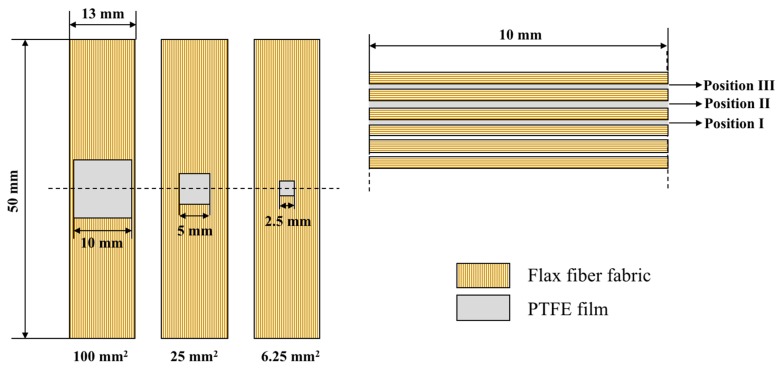
Schematic showing the location of the delamination introduced in the samples, and the respective identifications.

**Figure 2 materials-12-02559-f002:**
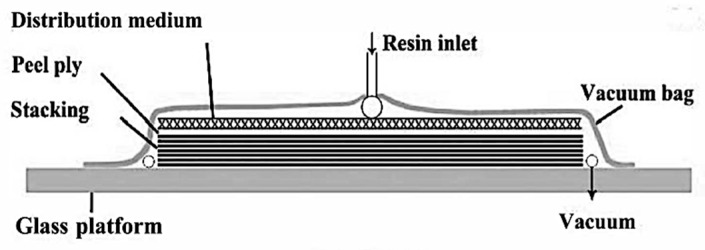
Schematic drawing of Vacuum-Assisted Resin Infusion (VARI) fabrication process for composite laminates.

**Figure 3 materials-12-02559-f003:**
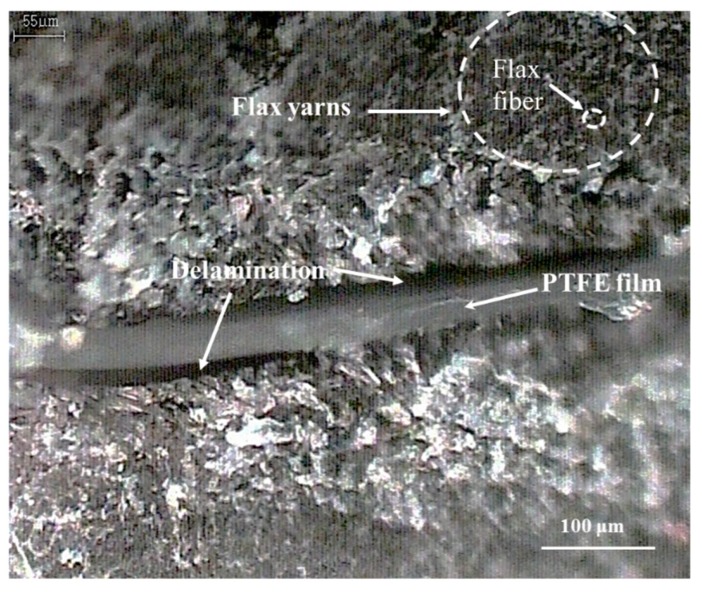
Cross-section micrographs of introduced delamination in the composite.

**Figure 4 materials-12-02559-f004:**
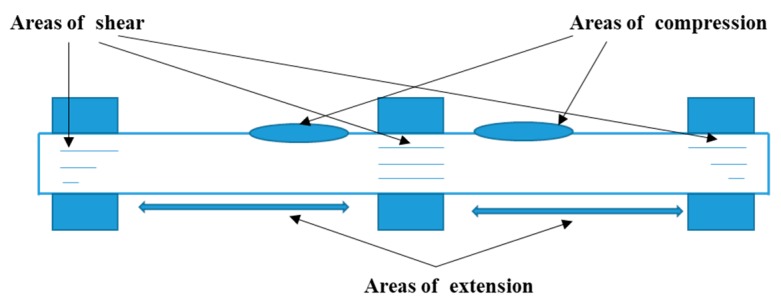
Schematic diagram of flexure test fixtures.

**Figure 5 materials-12-02559-f005:**
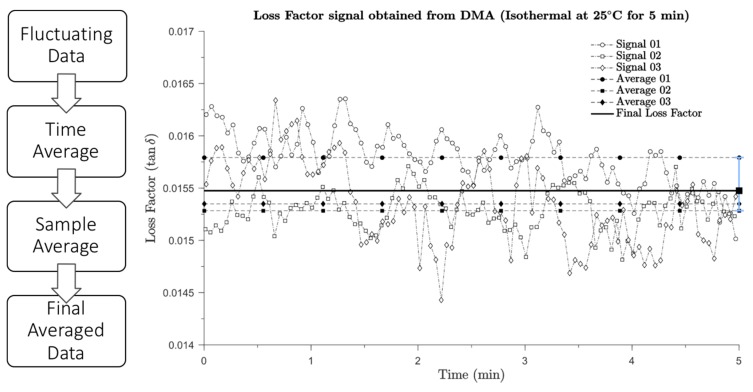
Illustration of the data processing used for the loss factor of specimen DEL_2.5_I at 1 Hz.

**Figure 6 materials-12-02559-f006:**
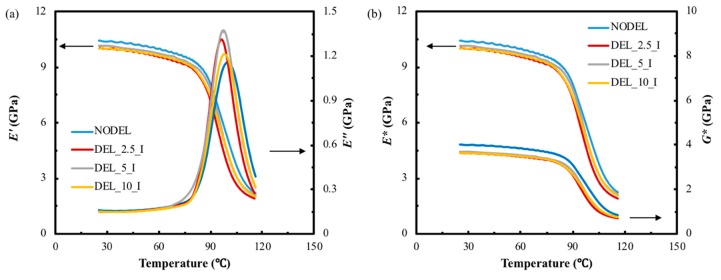
(**a**) Typical storage and lost moduli against temperature curves and (**b**) typical complex storage and shear moduli against temperature curves of flax fiber-reinforced polymer (FFRP_ composite with different delamination areas.

**Figure 7 materials-12-02559-f007:**
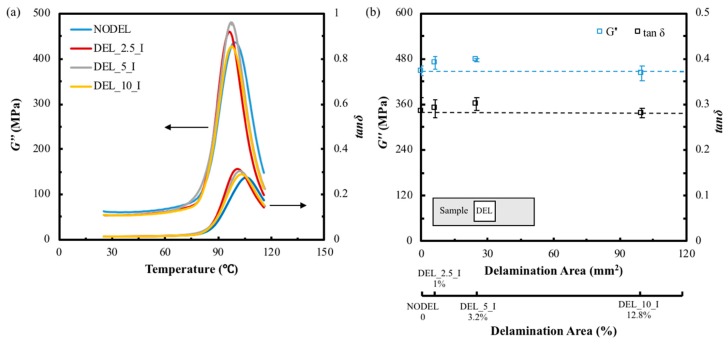
(**a**) Typical loss shear modulus and tanδ against temperature curves, (**b**) the loss shear modulus and loss factor of FFRP composites with different delamination areas.

**Figure 8 materials-12-02559-f008:**
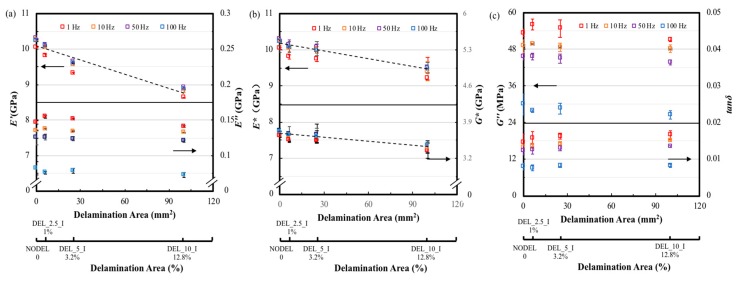
The dynamic mechanical properties of FFRP with different delamination areas tested at 25 °C, frequencies of 1, 10, 50 and 100 Hz, amplitude of 10 µm. Variation of (**a**) *E′* and *E″*, (**b**) *E** and *G** and (**c**) tanδ with delamination areas.

**Figure 9 materials-12-02559-f009:**
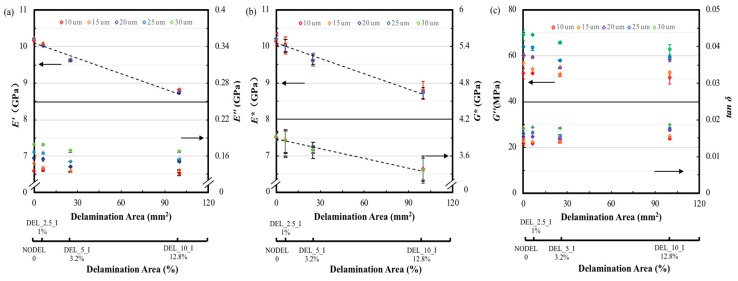
The dynamic mechanical properties of FFRP with different delamination areas tested at 25 °C, frequencies of 10 Hz, amplitude of 10, 15, 20, 25 and 30 µm. Variation of (**a**) *E′* and *E″*, (**b**) E* and *G** and (**c**) tanδ with delamination areas.

**Figure 10 materials-12-02559-f010:**
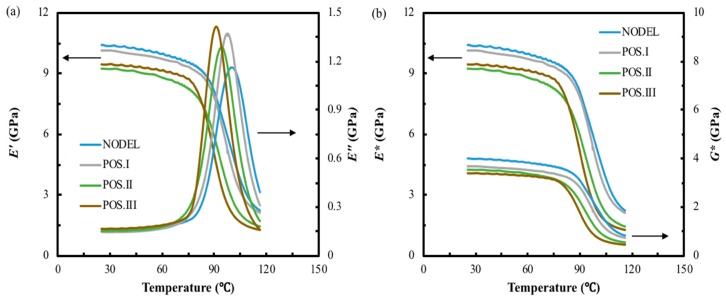
(**a**) Typical storage and lost moduli against temperature curves and (**b**) typical complex storage and shear moduli against temperature curves of FFRP composite with different transverse positions.

**Figure 11 materials-12-02559-f011:**
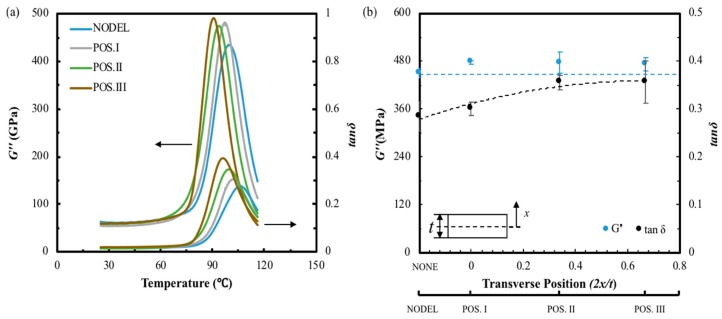
(**a**) Typical loss shear modulus and tanδ against temperature plots, (**b**) the loss shear modulus and loss factor of FFRP composites with different transverse position.

**Figure 12 materials-12-02559-f012:**
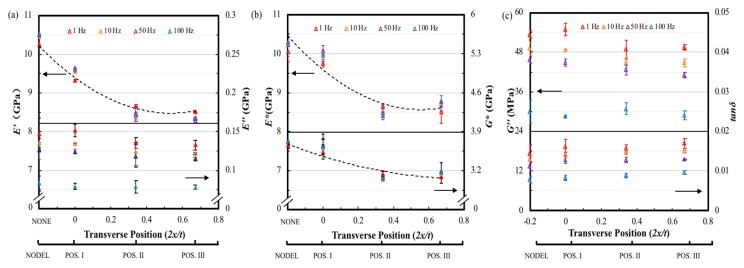
The dynamic mechanical properties of FFRP with different transverse delamination position tested at 25 °C, frequencies of 1, 10, 50 and 100 Hz, amplitude of 10 µm. Variation of (**a**) *E′* and *E″*, (**b**) *E** and *G** and (**c**) tanδ with transverse delamination position.

**Figure 13 materials-12-02559-f013:**
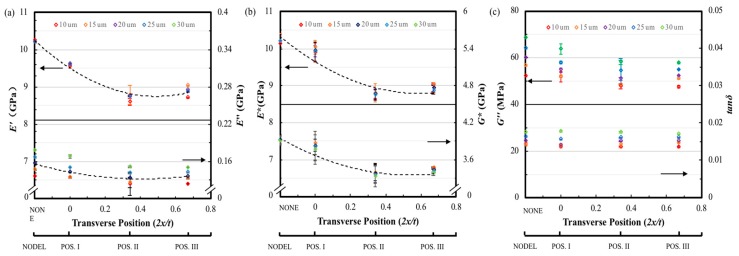
The dynamic mechanical properties of FFRP with different transverse delamination position tested at 25 °C, frequencies of 10 Hz, amplitude of 10, 15, 20, 25, and 30 µm. Variation of (**a**) *E′* and *E′*, (**b**) *E** and *G** and (**c**) tanδ with transverse delamination position.

**Table 1 materials-12-02559-t001:** Summary of delamination properties for different samples.

Sample ID	Dimensions (mm^3^)	Fiber Volume Fraction (%)	Delamination Area (mm^2^)	Delamination Position
NODEL	50 × 13 × 1.3	40	–	–
DEL_2.5_I			6.25	I
DEL_5_I			25	I
DEL_10_I			100	I
DEL_5_II			25	II
DEL_5_III			25	III

**Table 2 materials-12-02559-t002:** The test parameters setting of temperature.

Scanning Manner	Temperature (°C)	Ramp Rate (°C/min)	Frequency (Hz)	Amplitude (μm)
Temperature scans	25–120	5	10	10
Frequency scans	25	/	1, 10, 50, 100	10
Amplitude scans	25	/	10	10, 15, 20, 25, 30

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
