# Peer review of "Dynamic Mechanical Analysis on Delaminated Flax Fiber Reinforced Composites"

_materials, 2019, doi:10.3390/ma12162559_

Round 1
Reviewer 1 Report
In the paper “Dynamic mechanical analysis on delaminated flax fiber reinforced composite composites” the authors presents a new methodology for detecting delamination in composite laminate based on investigate its dynamic mechanical properties.
From my point of view, the topic of the present paper could be interesting for the readers of Materials . Nevertheless, I suggest the publication after minor revisions:
English needs improvement for example:
- Page 1 verse 34: transpiration interior parts it may be automotive interior parts
- Page 6 verse 210: The preset layering destroys the integration of laminates this sentence is incomprehensible and should be corrected
Other comments:
- How the authors chose the post-curing temperature (120°C for 2 h).
- How the authors determined fiber weight fractions
- The novelty of the proposed methodology should be more emphasized in relation to other commercial detection methods, e.g. ultrasonic and vibration methods
- what is the cost of using this method
- how samples for testing should be prepared
Reviewer 2 Report
This paper presents an interesting method to detect delamination defects in the composite materials made using flax fibers. Overall, this paper needs detailed proofreading as there are numerous grammatical mistakes, making it difficult to read. Some of the issues are mentioned below:
1. Line 18: This sentence is grammatically incorrect.
2. Line 22: replace Increasement by Increase
3. Line 41: mention which mechanical properties are you referring.
4. Line 67: DMA cannot characterize the molecular structure. Please mention the correct facts.
5. Line 66-69: Please add a reference to this line.
5. Line 86: Spelling mistake "Deamination".
6. Line 119: How did you measure the fiber vol. fraction?
7. Line 411: Spelling error. It is Transverse
8. Did you compare the found results with other techniques?
9. how does the sample with no defect compare to the samples with defects?
Reviewer 3 Report
Very interesting and clearly presented work and its concept.
Some language/formatting mistakes can be rarely found, e.g. :
page 2 line 86 should be "delamination";
page 2 line 88 "rese" - ?
page 3 line 100 "Unidirectional (UD)" because UD you use in line 108 without explaining “UD”
Round 2
Reviewer 2 Report
The authors have addressed all the concerns I had. The manuscript should be accepted for publication.